# Subgame Solving in Adversarial Team Games

**Brian Hu Zhang**[*]
Computer Science Department
Carnegie Mellon University
bhzhang@cs.cmu.edu

**Luca Carminati**[*]
DEIB, Politecnico di Milano
luca.carminat@polimi.it

**Federico Cacciamani**
DEIB, Politecnico di Milano
federico.cacciamani@polimi.it

**Gabriele Farina**
Computer Science Department
Carnegie Mellon University
gfarina@cs.cmu.edu

**Pierriccardo Olivieri**
DEIB, Politecnico di Milano
pierriccardo.olivieri@polimi.it

**Nicola Gatti**
DEIB, Politecnico di Milano
nicola.gatti@polimi.it

**Tuomas Sandholm**
Computer Science Department, CMU
Strategic Machine, Inc.
Strategy Robot, Inc.
Optimized Markets, Inc.
sandholm@cs.cmu.edu

## Abstract

In *adversarial team games*, a team of players sequentially faces a team of adversaries. These games are the simplest setting with multiple players where *cooperation* and *competition* coexist, and it is known that the information asymmetry among the team members makes equilibrium approximation computationally hard. Although much effort has been spent designing scalable algorithms, the problem of solving large game instances is open. In this paper, we extend the successful approach of solving huge two-*player* zero-sum games, where a blueprint strategy is computed *offline* by using an abstract version of the game and then it is refined *online*, that is, during a playthrough. In particular, to the best of our knowledge, our paper provides the first method for online strategy refinement via subgame solving in adversarial team games. Our method, based on the team belief DAG, generates a *gadget game* and then refine the blueprint strategy by using column-generation approaches in anytime fashion. If the blueprint is sparse, then our whole algorithm runs end-to-end in polynomial time given a best-response oracle; in particular, it avoids expanding the whole team belief DAG, which has exponential worst-case size. We apply our method to a standard test suite, and we empirically show the performance improvement of the strategies thanks to subgame solving.

## 1 Introduction

Sequential strategic games with multiple players where *cooperation* and *competition* coexist are fascinating problems receiving increasing interest in the scientific literature. The simplest case

---

[*]equal contribution; author order randomized

36th Conference on Neural Information Processing Systems (NeurIPS 2022).

is represented by *adversarial team games* (ATGs) [19], where a team of players faces a team of adversaries. This setting extends the basic zero-sum two-player setting, introducing the problem of coordinating team members with asymmetric and partial information. A common approach is that of *ex ante coordination* introduced in [7], in which the players can coordinate their strategies beforehand, but they have no means of communication during the gameplay. Many recreational and real-world scenarios can be modeled according to this framework, such as, *e.g.*, contract bridge, collusion in Poker, and sequential security games. While *ex ante* coordination makes the optimization problem convex, the problem of finding an equilibrium is inapproximable in polynomial time [7], and many efforts have been spent to design algorithms scaling up to non-toy instances.

**Related works.** The mainstream literature on ATGs, *e.g.*, [7, 8, 9, 23], focuses on *n-vs.-1* scenarios and resorts to column-generation (CG) algorithms to solve small/medium-sized instances exploiting the existence of equilibria with small support. These algorithms iterate between the problem of finding an equilibrium with a restricted set of strategies (the *meta-problem*) and the problem of enlarging such a set by finding the players' best responses. A crucial issue concerns the design of practical representations for the team's strategy space, which in the worst case is exponential in the size of the instance. In particular, a junction-tree-based decomposition is proposed in [20], while the team is represented through an explicit coordinator prescribing actions to team members in [5, 6]. The *team belief DAG* (TB-DAG) representation proposed in [22] bridges the previous two representations [5, 6, 20]. TB-DAG consists of a directed acyclic graph representation of the team strategy space, and it captures the players' beliefs over the game nodes given the joint strategy computed beforehand and the information observed publicly by the team players. A central concept in TB-DAG is *team belief* (or *team-public state*), which is a combination of the team players' information sets with the same public information. The size of TB-DAG is exponential in the number of information sets of each team-public state, and this number is called *information complexity* [22]. Counterfactual regret minimization (CFR) [24] and CG algorithms [14] applied to TB-DAG are the current state of the art to solve ATGs [21, 22]. Empirically, CFR outperforms CG when the information complexity is low, while CG outperforms CFR otherwise. However, despite the efforts to enhance representations and algorithms, solving medium-to-big-sized games is not currently affordable due to infeasible memory requirements for CFR and iterations requirements for CG [21].

The problem of scaling up to huge games in the two-player zero-sum setting has been successfully addressed by refining strategies *online* by subgame solving [1, 3, 4, 17]. This approach led to, *e.g.*, superhuman performances in poker [2, 16]. However, these techniques cannot be directly applied to ATGs due to some open non-trivial technical questions, such as, *e.g.*, how to generate the *gadget* game for the refinement, how to compute the counterfactual reach and best response value.

**Original contributions.** To the best of our knowledge, this paper provides the first method for refining strategies by subgame solving in ATGs. In particular, we generate a blueprint strategy by running a CG algorithm with a time limit. This procedure results in a blueprint whose support has a size linear in the number of iterations of the CG algorithm. We conveniently represent the blueprint by TB-DAG, while we take inspiration from maxmargin to design our algorithm to refine the blueprint strategy during the gameplay [15]. In particular, the gadget game is generated from the subgame of the TB-DAG played by strictly positive probability in the blueprint rooted in the players' public state in which the players are playing. Then, the CG algorithm is applied to the gadget subgame to refine the blueprint strategy with a time limit depending on the application. Here, the CG algorithm iteratively expands the gadget game by adding a new best response at any iteration.

By using CG for both the blueprint and the subgame solving procedures, the strategy is guaranteed to remain *sparse*. More precisely, the size of the representation of the strategy will grow only linearly in the number of best-response oracle (BRO) calls and polynomially in the size of the game. Hence, our algorithm is time- and space-efficient modulo the BRO, and in particular it does not need to expand the whole TB-DAG, which has worst-case exponential size. We hope that, in the future, this property will allow scaling to huge games.

The experimental evaluation of our method with a standard test suit shows that they dramatically reduce the gap between the values provided by the equilibrium strategy and the blueprint. In particular, the average reduction is of about 39% when the time limit used for the strategy refinement after every single move in the game equals the time needed by the CG algorithm to complete even a single iteration at the root of the game, and allotting more iterations further improves performance. Such an

empirical result shows that a local reoptimization can effectively lead to strategies whose value is close to the equilibrium value in ATGs.

## 2 Preliminaries

### 2.1 Adversarial team games

A *two-team extensive-form adversarial team game* (ATG) $\Gamma$, between teams ▲ and ▼, consists of:

- a directed tree of *histories*, or *nodes*, $\mathcal{H}$, whose edges are labeled with *actions*. The set of actions available at a node $h \in \mathcal{H}$ is denoted with $A(h)$, while the root of $\mathcal{H}$ is denoted with $\varnothing$. Leaves of the tree are called *terminal nodes*, their set is denoted with $\mathcal{Z}$. Given two nodes $h, h' \in \mathcal{H}$, we write $h \preceq h'$ if there exists a path in the tree going from $h$ to $h'$, and $h \prec h'$ if $h \preceq h'$ and $h \neq h'$;
- a partition $\mathcal{H} \setminus \mathcal{Z} = \mathcal{H}_{\mathsf{N}} \sqcup \mathcal{H}_{▲} \sqcup \mathcal{H}_{▼}$ of the set of nonterminal nodes determining who plays at each node: nature (N)—also called *chance*—, the max-team (▲), or the min-team (▼). Given a team $i \in \{▲, ▼\}$, the opponent is denoted with $-i$, and we let $\mathcal{H}_{-i} = \mathcal{H} \setminus \mathcal{H}_i$;
- for each team $i \in \{▲, ▼\}$, a partition $\mathcal{I}_i$ of $\mathcal{H}_i$ into *information sets*. In each information set $I \in \mathcal{I}_i$, every node in $I$ must have the same action set, denoted as $A(I)$;
- a *utility function* $u \in \mathbb{R}^{\mathcal{Z}}$ where $u[z]$ is the utility for ▲ for reaching terminal node $z$. Since the interaction between the teams is zero-sum, we let ▼'s utility be $-u[z]$;
- for each nature node $h \in \mathcal{H}_{\mathsf{N}}$, a distribution $p(\cdot|h)$ over the children of $h$. We will use $p[h]$ to denote the probability that nature plays *all* actions on the $\varnothing \to h$ path.

Since each $i \in \{▲, ▼\}$ is a *team* of players with potentially asymmetric information, we will *not* demand that the *meta* players, each representing a whole team, have perfect recall. However, for most of the paper, we will be interested in the special case where ▼ consists of just one player—that is, ▼ has perfect recall, but ▲ may not. In this paper, we will have no need whatsoever to differentiate between the *players* on the same team—rather, we will take the perspective of the team as a whole. Equivalently, this whole paper can be formulated in terms of *two-player games of imperfect recall*.

We will assume *timeability*—that is, no information set spans multiple levels of the tree [10]. Intuitively, this means that there is a turn clock in the game that is common knowledge to all players.

A *pure strategy* for team $i$ is a selection of one action for each information set $I \in \mathcal{I}_i$. The *realization form* $\boldsymbol{x}$ of a pure strategy is the vector $\boldsymbol{x} \in \mathbb{R}^{\mathcal{Z}}$ where $x[z] = 1$ if and only if the team plays every action on the path to $z$. A *correlated strategy* is a probability distribution over pure strategies, and its realization form is the appropriate convex combination. By taking arbitrary probability distributions over pure strategies, the individual players of a team can *correlate* their strategy with shared randomness hidden from the opposing team. The *best response* of a team $i$ is one of its strategies maximizing its expected utility given a fixed strategy of the opponent team $-i$.

The central solution concept in ATGs is the *Team-Maxmin Equilibrium with correlation* (TMEcor), introduced in [7]. A TMEcor is a pair of correlated strategies, one per team, such that each team's correlated strategy is a best response to the opponent's.

### 2.2 Public information and the team belief DAG

We call a piece of information *public* if it is common knowledge to all players. Formally, two nodes $h, h'$ in the same level of the tree are *indistinguishable to team $i$* if there is some information set $I \in \mathcal{I}_i$ that connects a descendant of $h$ to a descendant of $h'$. A *public state* is a connected component of the graph induced by the indistinguishability relation of both teams. A *team-public state* for team $i$ is a connected component of the graph given by the indistinguishability relation of that team alone.

Zhang et al. [22] introduce the *team belief DAG* (TB-DAG), a representation of the decision problem faced by the team in a team game. For a team $i$, the nodes $\mathcal{S}_i$ of the TB-DAG are partitioned in a set of *observation nodes* $\mathcal{O}_i$, *i.e.,* nodes in which the team observes information concerning the state of the game and a set of decision points—called *beliefs*—*i.e.*, nodes in which the team performs actions. To disambiguate the notation, the sets of beliefs for teams ▲ and ▼ are denoted with $\mathcal{B}$ and $\mathcal{C}$, respectively. Intuitively, both beliefs and observation nodes identify a subset of $\mathcal{H}$ that is indistinguishable to

the team, based on the common information they have. In what follows, we introduce the structure and basic notation of ▲'s TB-DAG representation. The representation of ▼'s TB-DAG is defined identically. The DAG is constructed recursively, starting from a root decision node that contains only the root node $\varnothing$ of the ATG and alternating decision and observation nodes along every possible path. More in detail, each edge leaving a belief $B$ corresponds to a so-called *prescription*, *i.e.,* a tuple that assigns an action to each infoset that has a nonempty intersection with $B$. Formally, the set of possible prescriptions at a belief $B$ is defined as $A(B) = \{\boldsymbol{a} \mid \boldsymbol{a} \in \times_{I_j \sim B} A(I_j)\}$, where, for any infoset $I \in \mathcal{I}_{\blacktriangle}$ and belief $B \in \mathcal{B}$, we write $I \sim B$ if $I \cap B \neq \varnothing$. Following a prescription $\boldsymbol{a}$ at a belief $B \in \mathcal{B}$, the TB-DAG transitions to the observation node are identified by

$$B\boldsymbol{a} = \bigcup_{\substack{I_j \in B \\ a_j \in \boldsymbol{a}}} \{ha_j \mid h \in I_j \cap B\} \cup \{ha \mid h \in B \cap \mathcal{H}_{-\blacktriangle}, a \in A(h)\} \in \mathcal{O}_{\blacktriangle}.$$

A belief $B \in \mathcal{B}$ is said to be terminal if it contains terminal nodes of the ATG. Finally, the set of possible observations that team can obtain from an observation node $O \in \mathcal{O}_{\blacktriangle}$ coincides with the set of ▲'s public states with nonempty intersection with $O$. Following the observation of a team public state $P$ at observation node $O$, the DAG transitions to the decision node identified by $O \cap P$. Let us remark that this construction ensures that each belief is a subset of a single team-public state. We denote ▲'s public state associated with belief $B \in \mathcal{B}$ as $\mathcal{P}(B)$, while the set of beliefs in a public state $P$ is denoted as $\mathcal{B}[P] = \{B \mid B \in \mathcal{B} \wedge B \in P\}$. Given a public state $P$ and a belief $B$, we say $B \preceq P$ if for any $h \in B$ there exists $h' \in P$ such that $h \preceq h'$.

For a TB-DAG, the strategy representation that we rely on is the *TB-DAG form*, that yields a strategy representation equivalent, yet more compact, to the set of correlated strategies. Given a pure strategy for a team ▲, the associated TB-DAG form is a vector $\boldsymbol{x} \in \{0, 1\}^{\mathcal{S}_{\blacktriangle}}$ such that $x[B] = 1$ if and only if the pure strategy prescribes to play all the actions from $\varnothing$ to all the nodes $h \in B$ and no $B' \supset B$ satisfying such property exists. Furthermore, for each observation node $B\boldsymbol{a} \in \mathcal{O}_{\blacktriangle}$, $x[B\boldsymbol{a}] = 1$ if and only if $x[B] = 1$ and the pure strategy prescribes to play all the actions $a_j \in \boldsymbol{a}$ with probability 1. The TB-DAG form of a correlated strategy is obtained by taking the appropriate convex combination of pure strategies' TB-DAG forms. The polytopes of TB-DAG-form strategies for team ▲ and ▼ are denoted as $\mathcal{X}$ and $\mathcal{Y}$ respectively. Throughout this work, it will be useful to consider the restriction of $\mathcal{X}$ and $\mathcal{Y}$ on the set of decision and observation nodes following a public state $P$. We denote such sets as $\mathcal{X}_P$ and $\mathcal{Y}_P$. Formally, the set $\mathcal{X}_P$ is defined by the following linear constraints:

$$x[B] = \sum_{\boldsymbol{a} \in A(B)} x[B\boldsymbol{a}] \quad \text{for non-terminal } B \succeq P \quad,$$

$$x[B'] = \sum_{B\boldsymbol{a} \supseteq B'} x[B\boldsymbol{a}] \quad \text{for } B' \succ P.$$

The set $\mathcal{Y}_P$ is defined similarly. Note that in the definition of $\mathcal{X}_P$ and $\mathcal{Y}_P$ the strategy values at initial beliefs at $P$ are unconstrained. Additional constraints can be introduced by the subgame solving algorithm used and will be discussed in Section 3.

The TB-DAG has worst-case exponential size, and therefore algorithms that build it fully cannot scale past a certain point. The crucial property we will need for this paper, though, is that pure strategies have sparse representations in the TB-DAG. In particular, every pure strategy plays to at most one belief in each team-public state.

## 3 Subgame solving for adversarial team games: Team-Maxmargin

### 3.1 Maxmargin

The maxmargin algorithm [15] is used in two-player zero-sum games in extensive form for online refinement of a *blueprint strategy*[2] for the game. In particular, for every public state $P$ reached by the players during the playthrough, maxmargin considers fixed the blueprint strategy in the *trunk* (*i.e.,* the part of the game leading to $P$), optimizing the player's strategy exclusively in the subgame rooted at $P$. Its main objective, roughly speaking, is to compute the strategy giving the largest decrease in

---

[2]A blueprint strategy is a suboptimal strategy for the whole game being computed on an abstracted, smaller version of the game.

exploitability when merged with the trunk one. For the sake of presentation, from here on, we refer to ▲ as the team player whose strategy is to be refined, and ▼ as the *opponent*. Furthermore, in this section, ▲ team player is composed of a single member.

The maxmargin algorithm works with an auxiliary game called the *gadget game*. The gadget game is obtained by adding additional nodes to the top of the subgame starting at $P$. The root of the gadget game is a newly-added ▼-node where the available moves—called *deviations*—allow him to choose an infoset among those belonging to $P$. Therefore, ▼ can adversarially decide from which of his own infosets the subgame will restart. Each of those actions leads to a chance node, where chance chooses a specific history in the infoset chosen by the opponent, based on chance reach probabilities and on the trunk blueprint strategy for ▲. Once a history has been selected by chance, the following part of the gadget game is the same as the original subgame up to the terminal nodes. Furthermore, the payoffs associated to each terminal node following a specific initial choice of infoset by the opponent are decreased by the *counterfactual best response value* at that infoset. This value is the best response value of ▼ conditioned on the game reaching that specific infoset, assuming that ▲ is playing the blueprint. The change of the payoffs induces ▼ to restart the gadget game in the infoset in which the current strategy of the refining player is most exploitable. Fully solving the gadget game guarantees *safety*: if maxmargin subgame solving is applied during a playthrough, then the resulting strategy is at most as exploitable as the blueprint.

The crucial issue when applying the maxmargin algorithm to an ATG is that the TB-DAG allows the same history to be reached through multiple beliefs, a phenomenon unique to team games. In particular, this issue will mean that it is difficult to formulate team subgame solving in terms of a gadget game the same way as is done in the standard two-player case. While this issue could be trivially solved by unrolling the TB-DAG into an extensive-form game [5, 6], such an unrolling is inefficient as it would require an amount of extra space exponential in the original game size [5, 6, 22]. We will therefore formulate the gadget "game" directly in terms of a max-min optimization problem that does not immediately correspond to an EFG.

## 3.2 Team-maxmargin

We introduce the *team-maxmargin* algorithm, which extends maxmargin subgame solving to ATGs. The algorithm that we present here can be applied to any ATG, without any constraint on the number of players in each team.

As aforementioned, in order to successfully apply subgame solving techniques to the TB-DAG representation of ATGs, there is the need to relate the TB-DAG-form polytopes of the two teams. To do so, we rebalance the reach of ▼'s strategy depending on the fixed trunk strategy of ▲, and nature. The normalization factor that we use is denoted as *counterfactual reach*.

**Definition 1** (Counterfactual reach). Let $\boldsymbol{x}$ the strategy for team ▲. For each ▼'s belief $C$, the counterfactual reach $\rho[\boldsymbol{x}, C]$ is the total nature and ▲ reach at $C$, defined as follows:

$$\rho[\boldsymbol{x}, C] := \sum_{h \in C} p[h] \sum_{B \ni h} x[B].$$

Counterfactual reach corresponds to the probability that ▲ and nature play to reach a specific belief $C$ given that ▼ plays a strategy $\boldsymbol{y}$ such that $y[C] = 1$. In the definition above, the counterfactual reach $\rho[\boldsymbol{x}, C]$ aggregates the realization probability of $\boldsymbol{x}$ and $p$ on the histories $h \in C$ so as to effectively represent the behavior of nature and ▲ in the polytope of the opponents.

The counterfactual reach allows us to determine which of ▼'s beliefs are actually reachable given ▲'s strategy. This set of nodes is denoted as the set of *counterfactually reachable beliefs*.

**Definition 2** (Counterfactually reachable beliefs). The set of all counterfactually reachable beliefs in a public state $P$ is denoted with $\mathcal{C}[\boldsymbol{x}, P]$. Formally:

$$\mathcal{C}[\boldsymbol{x}, P] := \{C \mid C \in \mathcal{C} \cap P \wedge \rho[\boldsymbol{x}, C] > 0\}.$$

Another concept needed to extend maxmargin to TB-DAGs is the one of *counterfactual best response value* at a ▼'s belief $C$, that is defined as the value of the best ▼'s prescription at $C$.

**Definition 3** (Counterfactual best response value). Let $\boldsymbol{x}$ be the strategy for team ▲. For each ▼ decision node $C$, let $u[\boldsymbol{x}, C]$ be the unnormalized counterfactual ▼-best response value defined by

the recurrences as follows:

$$u[\boldsymbol{x}, C] = \begin{cases} \displaystyle\sum_{z \in C} x[z]\,p[z]\,u[z] & \text{if } C \text{ is terminal} \\ \displaystyle\min_{\boldsymbol{a} \in A(C)} u[\boldsymbol{x}, C\boldsymbol{a}] & \text{otherwise} \end{cases},$$

$$u[\boldsymbol{x}, C\boldsymbol{a}] = \sum_{C' \subseteq C\boldsymbol{a}} u[\boldsymbol{x}, C'],$$

whereas the normalized counterfactual ▼-best response value at $C$ against $\boldsymbol{x}$ can be defined as follows:

$$u_N[\boldsymbol{x}, C\boldsymbol{a}] = \frac{u[\boldsymbol{x}, C\boldsymbol{a}]}{\rho[\boldsymbol{x}, C]}.$$

Notice that, in the definition of the counterfactual best response value, the realization form induced by the TB-DAG form strategy $\boldsymbol{x}$ and chance is taken into account. On the other hand, when considering the normalized counterfactual best response value, we condition to reaching a specific $C$. Hence, to remove the contribution to realizations due to actions that have been played in trunk, we divide by the counterfactual reach. Given the above definitions, we can now provide the optimization problem which the Team-Maxmargin algorithm needs to solve for the refinement of the strategy.

**Definition 4** (Team-maxmargin linear program). The maxmargin optimization problem in a TB-DAG subgame rooted at $P$ is formulated as the following linear program:

$$\max_{\boldsymbol{x}'} \min_{\boldsymbol{y}, \bar{\boldsymbol{y}}} \sum_{z \succeq P} x'[z]\,y[z]\,p[z]\,u[z] - \sum_{C \in \mathcal{C}[\boldsymbol{x}, P]} y[C]\,u[\boldsymbol{x}, C] \tag{1}$$

$$\text{s.t.} \qquad x'[B] = x[B] \qquad\qquad \text{for all ▲-beliefs } B \in \mathcal{B}[P] \tag{2}$$

$$y[C] = \frac{\bar{y}[C]}{\rho[\boldsymbol{x}, C]} \qquad\qquad \text{for all ▼-beliefs } C \in \mathcal{C}[\boldsymbol{x}, P] \tag{3}$$

$$\boldsymbol{x}' \in \mathcal{X}_P,\ \boldsymbol{y} \in \mathcal{Y}_P,\ \bar{\boldsymbol{y}} \in \Delta^{\mathcal{C}[\boldsymbol{x}, P]} \tag{4}$$

As in the original two-player formulation, ▼ can choose to deviate to any of the counterfactually reachable beliefs at the public state of the subgame by choosing $\bar{\boldsymbol{y}} \in \Delta^{\mathcal{C}[\boldsymbol{x}, P]}$. Equation 1 specifies the maximization of the *margin*, *i.e.*, the difference between the expected value of the subgame and the value of the counterfactual best response to ▲'s blueprint. This encodes the objective of ▲ to find a $\boldsymbol{x}'$ to maximize its value w.r.t. the blueprint for any adversarially chosen deviation of the opponent. Equation 2 fixes the trunk strategy of ▲. Equation 3 rescales reach probabilities of the opponent according to the counterfactual reach $\rho[\boldsymbol{x}, C]$. This is equivalent to normalizing by the nature and ▲ reach probabilities at $C$. Such a normalization is crucial because it accounts for the fixed reach probabilities in the trunk for chance and ▲. While it may seem more intuitive to normalize by the reach probabilities of ▲ and chance directly on the terminal nodes, as is typically done in two-player zero-sum games, in team games it is actually necessary to do so on the beliefs in $\mathcal{C}[\boldsymbol{x}, P]$. This a consequence of the fact that, when ▼ is composed by more than one player, the reach of ▲ and nature over terminal nodes depends directly on the deviation $C \in \mathcal{C}[\boldsymbol{x}, P]$ chosen by ▼, and multiple such beliefs $C$ can reach the same terminal node. This is a difficulty that only occurs in the team-vs-team setting. Note that only counterfactually reachable beliefs $\mathcal{C}[\boldsymbol{x}, P]$ of ▼ are considered in this constraint since they are the only potential deviations of the opponent in the gadget game. Equation 4 encompasses all the TB-DAG constraints on the strategy spaces of ▲ and ▼.

One may expect that, in parallel to the two-player case, our team-maxmargin optimization problem may also be representable as an ATG. However, the constraints that we have introduced into the formulation go out of the scope of ATGs, so this is not directly possible. Despite this, the strategy spaces of both players in Definition 4 are DAGs, to which any technique for solving team games, such as CFR [24] or column generation, apply.

A complete construction procedure of the gadget DAG is provided in Appendix A. Furthermore, in Appendix B, we extend other subgame solving techniques (*i.e.*, resolving [4] and reach [1]) to the team setting.

**Safety of the refinement.** The team maxmargin algorithm achieves the same safety guarantees as the maxmargin algorithm, for similar arguments to [15, 1]. Formally:

---

**Algorithm 1** Maxmargin subgame solving with column generation, at public state $P$

---

1: **Input:**
2:   sparse DAG-form blueprint $\boldsymbol{x}$
3:   reach probabilities $\rho[\boldsymbol{x}, C]$ and alt values $u[\boldsymbol{x}, C]$ for *all reachable* ▼-beliefs $C \in \mathcal{C}[\boldsymbol{x}, P]$
4: let $\tilde{\mathcal{B}}$ be the set of ▲-beliefs in $P$ with $x[B] > 0$.
5: $\boldsymbol{x}'_{B,1} \leftarrow \{\text{GENERATEARBITRARYSTRATEGY}(B)\}$ for each $B \in \tilde{\mathcal{B}}$
   ▷ *for each time $t$ and belief $B \in \tilde{\mathcal{B}}$, $\boldsymbol{x}'_{B,t}$ is a pure strategy defined on the sub-DAG rooted*
   ▷ *at $B$. It represents a strategy that ▲ may take following belief $B$.*
6: **for** iteration $t = 1, 2, \ldots$ **do**
7:   solve the *meta-game*:

$$\max_{\boldsymbol{\lambda}_B : B \in \tilde{\mathcal{B}}, \lambda_* \geq 0} \min_{\boldsymbol{y}, \bar{\boldsymbol{y}}} \sum_{z \succeq P} x'[z] y[z] p[z] u[z] - \sum_{C \in \mathcal{C}[P]} y[C] u[\boldsymbol{x}, C]$$

$$\text{s.t.} \qquad \boldsymbol{x}' = \lambda_* \boldsymbol{x} + \sum_{B \in \tilde{\mathcal{B}}} x[B] \sum_{1 \leq \tau \leq t} \lambda_{B,\tau} \boldsymbol{x}'_{B,\tau}$$

$$y[C] = \bar{y}[C] / \rho[\boldsymbol{x}, C] \quad \text{for all ▼-beliefs } C \in \mathcal{C}[\boldsymbol{x}, P]$$

$$\sum_{1 \leq \tau \leq t} \lambda_{B,\tau} = 1 - \lambda_* \qquad \text{for all ▲-beliefs } B \in \tilde{\mathcal{B}}$$

$$\boldsymbol{\lambda}_B \geq 0, \lambda_* \geq 0, \Delta^t, \ \boldsymbol{y} \in \mathcal{Y}_P, \ \bar{\boldsymbol{y}} \in \Delta^{\mathcal{C}[\boldsymbol{x}, P]}$$

8:   **for** each belief $B \in \tilde{\mathcal{B}}$ **do**
9:     solve for a *best response* to $\boldsymbol{y}$ given belief $B$:

$$\boldsymbol{x}'_{B,t+1} \leftarrow \operatorname*{argmax}_{\boldsymbol{x}' \in \{0,1\}^{\mathcal{H}_B}} \sum_{z \succeq B} x'[z] y[z] p[z] u[z]$$

$$\text{s.t.} \quad x'[ha] x'[h'] = x'[h'] x'[ha] \quad \forall h, h' \in I \in \mathcal{I}_{\blacktriangle} \tag{5}$$

$$x'[h] = 1 \qquad \forall h \in B$$

   where $\mathcal{H}_B$ is the set of nodes $h \succeq B$.
10: **return** $\boldsymbol{x}'$ converted to sparse TB-DAG form

---

**Theorem 5.** *Applying team maxmargin subgame solving (Definition 4) to every subgame reached during play, in a nested fashion, results in a playing a strategy with exploitability no worse than that of the blueprint.*

A proof of Theorem 5 can be found in Appendix D.

**Team-maxmargin algorithm.**   The team-maxmargin algorithm is applied similarly to the original maxmargin algorithm. If we want to locally refine a strategy $\boldsymbol{x}$ of ▲ while playing, team-maxmargin proceeds as follows: at each game turn, it considers the public state $P$ reached and computes an approximate solution $\boldsymbol{x}'$ to the linear program in Definition 4 (while respecting a timelimit). Then $\boldsymbol{x}$ is updated for the next turns: $x[B] \leftarrow x'[B] \ \forall B \in \mathcal{X}_P$. The blueprint strategy is used as the first strategy $\boldsymbol{x}$. We solve the linear program by means of a column generation procedure in which the current strategy $\boldsymbol{x}$ is given as an available choice. This guarantees that the solution found is not worse than the current strategy, even if a single iteration is performed.

## 4   Column generation for sparser solutions

In this section, we argue that, if the opponent ▼ is a single player (rather than a team) and the blueprint is sparse, then, by using column generation (*e.g.*, Farina et al. [8]) as the solver, we can arrive at an online game-playing algorithm that runs in polynomial time (in $\mathcal{H}$, per iteration) with the exception of a best-response oracle.

The key observation is that the maxmargin formulation of subgame solving (Definition 4) has an *input* of size $|\mathcal{B}| + |\mathcal{C}|$ where $\mathcal{B}$ is the set of *played* beliefs for ▲ in the current public state, and $\mathcal{C}$ is the set of *all* beliefs for ▼ in the current public state. Therefore, to have a hope of an efficient algorithm, we need both of these quantities to be small. Hence, for this section, let us assume that ▼ is a single player or a team whose DAG is small (so that $|\mathcal{C}|$ is small) and that the blueprint is sparse (so that $|\mathcal{B}|$ is small at least on the first iteration).

The algorithm is given in Algorithm 1. At a high level, for computing best responses, it splits the problem of solving a game starting at public state $P$ into subproblems, one for each reachable ▲-belief $B$. This allows a "finer" mixing of strategies than would have been allowed otherwise. Given a best-response oracle (*i.e.*, a solution method to the integer program (5)), the whole algorithm runs in time $\mathrm{poly}(|\mathcal{H}_P|, |\tilde{\mathcal{B}}|, |\mathcal{C}[P]|, t)$ where $\mathcal{H}_P$ is the set of nodes $h \succeq P$ and $\tilde{\mathcal{B}}$ is the set of ▲-beliefs in $P$ with positive reach. While NP-hard in the general case, integer programs are practically reasonably fast to solve. In particular, if a polynomial-space algorithm such as depth-first branch-and-bound is used to solve the integer program, then the whole algorithm runs in polynomial space. This is in stark contrast to any algorithm that would expand the whole TB-DAG, as the TB-DAG takes worst-case exponential space.

**The creation of sparse blueprints.**  Throughout this section, we have been assuming that blueprints created are sparse, in particular, that in every public state there are at most polynomially many beliefs. There are two straightforward ways to guarantee this. The first is to use a natively sparse algorithm such as column generation to generate the blueprint in the first place. The second is to take an arbitrary blueprint in realization form, and express it as a sparse convex combination via Caratheodory's theorem[3]. Then, the blueprint will have support size at most $|\mathcal{Z}|$. On the other hand, the support size of the blueprint generated by a CG algorithm, scales linearly with the number of iterations, which, under reasonable time constraints, rarely exceeds the hundreds. Thus, the far smaller support size of the blueprint yielded by CG is a point in favour of the adoption of the former family of algorithms for blueprint generation.

## 5  Experimental evaluation

### 5.1  Experimental setting

**Game instances.**  In our evaluation, we resort to game instances whose exact solution can be computed in practice by standard algorithms available in the literature. This choice allows us to calculate the equilibrium value, which is necessary to appropriately assess how our subgame solving method approximates the exact solution as the time spent for the strategy refinement increases. In particular, our test suite is composed of parametric versions of the ATG instances customarily adopted in the literature where the adversary team is composed of a single player: Kuhn [11] and Leduc [18] poker with team collusion, team Liar's Dice [13] and Tricks [21] (a simplified Bridge endgame.)

For space reasons, we omit the rules of these games, pointing an interested reader to the aforementioned articles. We use the following labels:

- K*nr*: *n*-player Kuhn poker with *r* ranks;
- L*nbrs*: *n*-player Leduc poker with *b* bets in each betting round, *r* ranks and *s* suits;
- D*nd*: *n*-player Liar's Dice with one *d*-sided dice per player;
- T*d*: three-player Trick-taking game with three ranks, four suits, and a limited amount of possible deals fixed to *d*.

We also use a binary string of length $n$ to denote the assignment of teams, where the $i$th character is 1 if and only if the $i$th player is on team ▲.

**Blueprint and strategy refinement time limits.**  As usual with subgame solving methods, the performance depends on the quality of the blueprint and the time spent for its refinement after every single move in the game. To provide coherent results over multiple, different game instances, we set

---

[3]Caratheodory's theorem asserts that any convex combination of points in $\mathbb{R}^d$ can be expressed as a convex combination of at most $d + 1$ of them.

Table 1: Team's values against a best-responding opponent when playing the equilibrium strategy, blueprint, or refinement strategy with $\alpha = 5$, and corresponding algorithms running times. When the equilibrium computation requires more than 2 hours, we use the equilibrium values provided in [21]. The table summarizes the experiments for different game instances (see Section 5.1) and team structures (the $i$-th element of the vector equal to 1 means that player $i$ is in team ▲), with $\alpha = 5$. Equilibrium time, refinement time (per move) and blueprint time denote, respectively, the time needed to find an exact equilibrium via CG, the time allocated to a single iteration of subgame solving and the time allotted to blueprint computation. Equilibrium value indicates the expected utility of the team at the equilibrium, while refinement value and blueprint value represent the team expected utility when they play against a best-responding opponent team and adopt, respectively, the blueprint strategy and the refined strategy. The gap reduction column quantifies the improvement of the refined strategy over the blueprint strategy (see Section 5.2 for a formal definition of *gap*).

| Game instance | Team structure | Equilibrium time | Refinement time (per move) | Blueprint time | Equilibrium value | Refinement value | Blueprint value | Gap reduction |
|---|---|---|---|---|---|---|---|---|
| K34 | 110 | 0.05s | 0.02s | 0.02s | -0.042 | -0.050 | -0.061 | 58.0% |
| K36 | 110 | 0.55s | 0.07s | 0.03s | -0.024 | -0.055 | -0.098 | 58.3% |
| K38 | 110 | 2.58s | 0.20s | 0.04s | -0.019 | -0.031 | -0.152 | 91.2% |
| K312 | 110 | 10.73s | 0.74s | 0.29s | -0.014 | -0.024 | -0.064 | 78.9% |
| K45 | 1110 | 6.92s | 0.54s | 0.22s | -0.030 | -0.046 | -0.354 | 95.2% |
| L3133 | 110 | 6m 39s | 0.57s | 1.59s | 0.215 | 0.038 | -0.616 | 78.7% |
| L3143 | 110 | >2h | 2.59s | 6.21s | 0.107 | -0.133 | -0.673 | 69.2% |
| L3151 | 110 | 2m 46s | 1.27s | 2.80s | -0.019 | -0.399 | -0.624 | 37.3% |
| L3153 | 110 | >2h | 4.91s | 8.84s | 0.024 | -0.177 | -0.681 | 71.5% |
| L3223 | 110 | 7m 44s | 1.02s | 12.28s | 0.516 | -0.320 | -1.299 | 54.0% |
| L3523 | 110 | >2h | 3m 21s | 10m 46s | 0.953 | -1.151 | -6.671 | 72.4% |
| D33 | 110 | 3m 55s | 1.06s | 9.15s | 0.284 | 0.279 | 0.238 | 90.2% |
| D34 | 110 | >2h | 33.21s | 10m 11s | 0.284 | 0.241 | 0.139 | 70.0% |
| D62 | 111110 | >2h | 40.22s | 10m 11s | 0.333 | 0.167 | -0.000 | 50.0% |
| T350 | 101 | >2h | 0.49s | 1.09s | 0.600 | 0.509 | 0.352 | 63.3% |
| T3100 | 101 | >2h | 1.16s | 1.85s | 0.710 | 0.514 | 0.495 | 8.5% |

both of these time limits dependent on the complexity of the game. More precisely, the blueprint computation is stopped once 10 minutes have elapsed or column generation has achieved a Nash gap of $\Delta/10$, where $\Delta$ is the difference between maximum and minimum team's payoffs, whichever comes first. We remark that the main goal of our experimental evaluation is to show that, no matter the suboptimality of the blueprint strategy and the amount of computation allocated to the subgame solving routine, the local re-optimization performed under our subgame solving framework allows us to improve the starting strategy. The choice of focusing on cases in which the blueprint strategy is suboptimal, and in which the computational budget available for subgame solve is (even severely) limited, goes precisely in that direction.

To create the strategy that would be played by Algorithm 1, we perform refinement at every public state, in a top-down order. We use a range of time limits for the strategy refinement, defined as the average time needed by a single iteration of the CG algorithm at the root of the whole game multiplied by a number $\alpha \in \{0, 1, ..., 10\}$. Notice that the value obtained with $\alpha = 0$ is the value provided by the blueprint, while using $\alpha \geq 1$ ensures that at least one iteration of strategy refinement is done after every move in the game. Each experiment was allocated 32 CPU cores and 256 GB RAM on a cluster machine. Integer and linear programs were solved with Gurobi 9.5.

## 5.2 Discussion of the results

For every game instance, we computed the best-response values for the opponent against the following strategies: the equilibrium strategy, the blueprint strategy, and the strategy returned by our refinement method for different values of $\alpha$. We will use *gap* to refer to the difference between a value and the equilibrium value. Intuitively, it shows the relative quality of the blueprint when compared with the equilibrium. We use a performance index based on the concept of gap showing the capability of our refinement method to reduce the gap. We call it *gap reduction*. It is defined as $1 - (E - R)/(E - B)$, where $E$, $R$, and $B$ are the values of the equilibrium, refined strategies, and blueprint respectively.

The values provided by the strategies when $\alpha = 5$ are provided in Table 1, together with the algorithms' runtimes. We initially observe that our experimental evaluation confirms the maxmargin theoretical results as the team's exploitability never increases. Most importantly, with the majority of the game instances, the gap reduction is dramatic (on the average, about 65%). These results are coherent with the results achieved in the literature for two-player zero-sum games [1, 15]. We also investigate how the performance of our strategy refinement method varies as $\alpha$ varies. In Figure 1, we report the results related to the two most representative instances together with their blueprint and equilibrium values. The plots and the tables related to the entire test suite are in Appendix C.

Interestingly, we observe that, with the vast majority of the game instances, the curve describing the improvement provides the maximum increase by the first iteration of strategy refinement (corresponding to $\alpha = 1$). In particular the average gap reduction is about 39% when $\alpha = 1$, showing that a time limit equal to the time needed for a single CG iteration at the root of the tree is sufficient to get a value very close to the equilibrium value. We also observe that the equilibrium value is never completely recovered in the test suite. This is not surprising as strategy refinement methods perform a local reoptimization. The improvement is sometimes not monotonically increasing in the number of the iterations. This is due to the nature of CG algorithms.

## 6  Conclusions

In this paper, we propose subgame solving algorithms for adversarial team games based on the team belief DAG. In team-vs.-player games, when applied with column generation as both the blueprint generator and subgame solving algorithm, our techniques run efficiently modulo a best-response oracle, which in practice can be implemented via an integer program solver. In general team-vs.-team games, our method addresses several technical difficulties that do not arise in the two-*player* setting, most notably the normalization of reach probabilities required by maxmargin. In the experimental activity, we find that applying any amount of subgame solving at all to a blueprint, even a single iteration of column generation, is significantly superior to not applying it. More precisely, with a standard test suite, our method reduces the the gap between the equilibrium value and the value provided by the blueprint by 39% on average even when $\alpha = 1$, and up to 65% when $\alpha = 5$.

In the future, we believe that our contributions can allow the scaling of subgame solving techniques to very large team games, where column-generation-like methods such as PSRO [12] will likely be the fundamental building block for practical solutions.

## Acknowledgements

This material is based on work supported by the National Science Foundation under grants IIS-1901403 and CCF-1733556, and the ARO under award W911NF2010081.

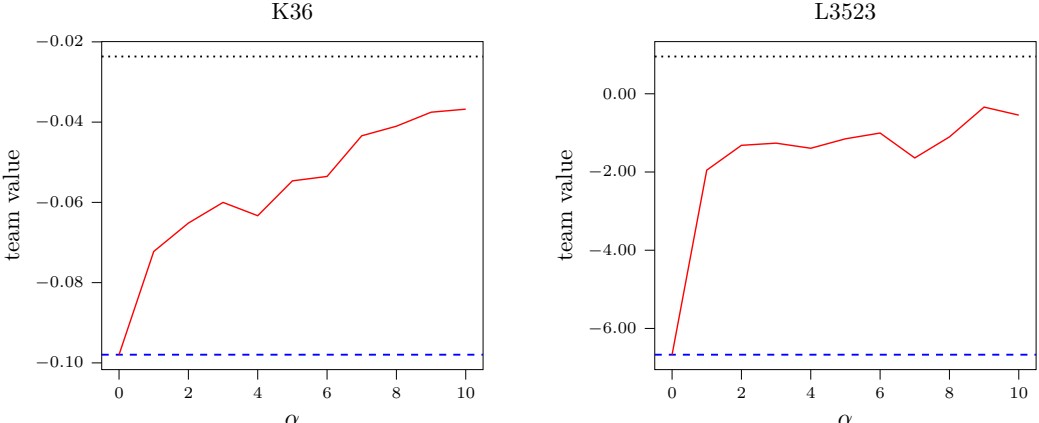

Figure 1: Value of the team's refined strategy as varying the refinement time limit. The blue dashed line in the bottom is the blueprint value, while the black dotted line in the top is the equilibrium value.

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
