---

**Algorithm 2** Construction of a Gadget DAG equivalent to Definition 4

---

1: **Input:**
2:   ▲-reach probabilities $x[B]$ on beliefs $B \in \mathcal{B}[P]$
3:   cf. reach probabilities $\rho[\boldsymbol{x}, C]$ and alt values $u[\boldsymbol{x}, C]$ for *all reachable* ▼-beliefs $C \in \mathcal{C}[\boldsymbol{x}, P]$
4:   already initialized DAGs $\mathcal{D}_{\blacktriangle}$ and $\mathcal{D}_{\blacktriangledown}$
5: let $\tilde{\mathcal{B}}$ be the set of ▲-beliefs in $P$ with $x[B] > 0$.
   ▷ *Considering $\mathcal{D}_{\blacktriangle}$*
6: $s_{\blacktriangle}^o \leftarrow$ new observation node in $\mathcal{D}_{\blacktriangle}$
7: $s_{\blacktriangle}^d \leftarrow$ new decision node in $\mathcal{D}_{\blacktriangle}$
8: add edge $s_{\blacktriangle}^o \to s_{\blacktriangle}^d$
9: **for** each belief $B \in \tilde{\mathcal{B}}$ **do**
10:   $s_{\blacktriangle}^B \leftarrow$ new observation node in $\mathcal{D}_{\blacktriangle}$
11:   add edge $s_{\blacktriangle}^d \to s_{\blacktriangle}^B$ **with fixed probability** $x[B]$
12:   add edge $s_{\blacktriangle}' \to B$
   ▷ *Considering $\mathcal{D}_{\blacktriangledown}$*
13: $s_{\blacktriangledown}^o \leftarrow$ new observation node in $\mathcal{D}_{\blacktriangledown}$
14: $s_{\blacktriangledown}^d \leftarrow$ new decision node in $\mathcal{D}_{\blacktriangledown}$
15: add edge $s_{\blacktriangledown}^o \to s_{\blacktriangledown}^d$
16: **for** each belief $C \in \mathcal{C}[\boldsymbol{x}, P]$ **do**
17:   $s_{\blacktriangledown}^C \leftarrow$ new observation node in $\mathcal{D}_{\blacktriangledown}$
18:   add edge $s_{\blacktriangledown}^d \to s_{\blacktriangledown}^C$
19:   add edge $s_{\blacktriangledown}' \to C$
20:   **associate a payoff to** $s_{\blacktriangledown}^C$ **equal to** $-u[\boldsymbol{x}, C]$
21:   **divide reach along edge** $s_{\blacktriangledown}^d \to s_{\blacktriangledown}^C$ **by** $\rho[\boldsymbol{x}, C]$
22: **return** the updated $\mathcal{D}_{\blacktriangle}$ and $\mathcal{D}_{\blacktriangledown}$

---

## A   Procedure for DAG gadget

Moravcík et al. [15] present a gadget game equivalent to maxmargin's linear program in the two-player setting. Similarly, it is possible to express the team-maxmargin linear program as an equivalent gadget DAG to solve. While the gadget DAG solution for the refining player corresponds to the solution given by the team-maxmargin linear program, there does not exist an ATG such that the associated TB-DAG is the gadget DAG. We will therefore avoid calling our gadget DAG a *game*, since a game with an identical strategy space does not exist.

The gadget DAG can be constructed from the TB-DAG of the subgame by considering, for both teams, the *root beliefs* (*i.e.* the beliefs at the root of the subgame to be refined), and all following nodes. In each team's DAG, we introduce a root observation node followed by a single decision node, which we call the *gadget belief*. Its actions lead as many observation nodes as belief considered at the root of the subgame. Each of those observation nodes will have a single observation, each leading to a different root belief.

In addition, some constraints and extra payoffs have to be introduced. For ▲'s gadget DAG, each root belief $B$ must be played with *unnormalized* "probability" $\boldsymbol{x}[B]$. (Note that $\sum x[B] \neq 1$ in general). For ▼'s gadget DAG, the reach associated to the observation nodes following the gadget belief is divided by a factor $\rho[\boldsymbol{x}, C]$, where $C$ is the belief they are reaching. Moreover, a payoff $u[\boldsymbol{x}, C]$ is associated to those observation nodes. Algorithm 2 presents a procedure corresponding to the described construction.

Such constraints are not directly representable in the TB-DAG formalism, since reaches may be greater than 1 (after division by $\rho[\boldsymbol{x}, C]$), and since there are decision nodes with fixed strategy and payoffs on observation nodes. These constraints cannot arise when constructing the TB-DAG of an AGT, and therefore no AGT corresponds to our gadget DAG. However, such constraints maintain the scaled extension structure as specified in [22], and therefore the same CFR algorithm used to solve DAGs can be used.

In practice for our experiments, we do not actually construct the DAG of ▲, since it is often too big. Instead, we use a column-generation algorithm to solve the subgame—see Algorithm 1. However,

in domains where ▲ has small information complexity (see Zhang et al. [22]), performing subgame solving in this manner on the original DAG (or a reasonable abstraction thereof) is possible.

## B  Other Subgame Solving techniques

### B.1  Gifts

Considering counterfactual best response values as the maximum counterfactual values allowed for the opponent in any of his root infosets is a sufficient condition to guarantee that exploitability of the refined strategy will not be higher than that of th e blueprint.

In [1], a relaxation of this condition is proposed. For each action $a$ of the opponent in a infoset, the *gift* is defined as the difference between the counterfactual best response value of the best action and the one of $a$. If an action has positive gift, that means that the action is suboptimal. Therefore, the refining player can increase the counterfactual best response value of any action having a positive gift without incurring any exploitability increase. In reach-maxmargin [1], the sum of gifts over actions played before reaching an infoset is used to increase the associated counterfactual best response values.

In the following, we will extend the definitions of gifts presented in [1] to team-maxmargin, creating a team-reach-maxmargin algorithm. Two possible definitions will be presented, as originally presented in [1]: the lower bound one gives stronger theoretical guarantees of exploitability reduction, while the safe one gives stronger empirical performance while still retaining safety.

**Definition 6** (Gifts). The gift associated to an action $a$ played in a belief $C$ is defined as:

$$\tilde{g}[\boldsymbol{x}, Ca] := \left( \min_{a'} u[\boldsymbol{x}, Ca'] - u[\boldsymbol{x}, Ca] \right) \frac{1}{\rho[\boldsymbol{x}, C]}.$$

The updated team-maxmargin objective can be formulated as:

$$\max_{\boldsymbol{x}'} \min_{\boldsymbol{y}, \bar{\boldsymbol{y}}} \sum_{z \succeq P} x'[z]\, y[z]\, p[z]\, u[z] - \sum_{C \in \mathcal{C}[P]} y[C] \left( u[\boldsymbol{x}, C] + \rho[\boldsymbol{x}, C] \min_{\pi \text{ path } \varnothing \to C} \sum_{C'a' \in \pi} g[\boldsymbol{x}, C'a'] \right).$$

The minimization over paths $\varnothing \to C$ is taken over ▼'s DAG, and represents the possibility that there can be multiple ways of reaching the same belief. Such a minimization can be computed efficiently (in the size of ▼'s DAG) via dynamic programming.

### B.2  Resolving

Similarly to maxmargin, resolving [4] can be formulated for team games as well, by taking Definition 4 and considering any strategy achieving positive margin for every ▼ belief.

**Definition 7** (Resolving linear program). Resolving optimization problem in a TB-DAG subgame rooted at $P$ can be expressed as the following linear program.

$$\max_{\boldsymbol{x}'}\ 0$$
$$\text{s.t.} \ \min_{\boldsymbol{y}, \bar{\boldsymbol{y}}} \sum_{z \succeq P} x'[z]\, y[z]\, p[z]\, u[z] - \sum_{C \in \mathcal{C}[P]} y[C] u[\boldsymbol{x}, C] \geq 0$$
$$x[B] = x'[B] \qquad \text{for all ▲-beliefs } B \in \mathcal{B}[P]$$
$$y[C] = \frac{\bar{y}[C]}{\rho[\boldsymbol{x}, C]} \qquad \text{for all ▼-beliefs } C \in \mathcal{C}[\boldsymbol{x}, P]$$
$$\boldsymbol{x}' \in \mathcal{X}_P,\ \boldsymbol{y} \in \mathcal{Y}_P,\ \bar{\boldsymbol{y}} \in \Delta^{\mathcal{C}[\boldsymbol{x}, P]}$$

## C  Complete experiment results on refinement time

In the following, we show the complete experimental results: a table as Table 1 but considering the specific case of $\alpha = 1$ (instead of $\alpha = 5$), and the plots of the value of the refined strategy against a best responding opponent, for varying $\alpha$ and for all games in our benchmark.

Table 2: Team's values against a best-responding opponent when playing the equilibrium strategy, blueprint, or refinement strategy with $\alpha = 1$, and corresponding algorithms running times. When the equilibrium computation requires more than 2 hours, we use the equilibrium values provided in [21]. The table summarizes the experiments for different game instances (see Section 5.1) and team structures (the $i$-th element of the vector equal to 1 means that player $i$ is in team ▲), with $\alpha = 5$. Equilibrium time, refinement time (per move) and blueprint time denote, respectively, the time needed to find an exact equilibrium via CG, the time allocated to a single iteration of subgame solving and the time allotted to blueprint computation. Equilibrium value indicates the expected utility of the team at the equilibrium, while refinement value and blueprint value represent the team expected utility when they play against a best-responding opponent team and adopt, respectively, the blueprint strategy and the refined strategy. The gap reduction column quantifies the improvement of the refined strategy over the blueprint strategy (see Section 5.2 for a formal definition of *gap*

| Game instance | Team structure | Equilibrium time | Refinement time (per move) | Blueprint time | Equilibrium value | Refinement value | Blueprint value | Gap reduction |
|---|---|---|---|---|---|---|---|---|
| K34 | 110 | 0.05s | 0.00s | 0.02s | -0.042 | -0.061 | -0.061 | 0.0% |
| K36 | 110 | 0.55s | 0.01s | 0.03s | -0.024 | -0.072 | -0.098 | 34.6% |
| K38 | 110 | 2.58s | 0.04s | 0.04s | -0.019 | -0.149 | -0.152 | 2.0% |
| K312 | 110 | 10.73s | 0.15s | 0.29s | -0.014 | -0.043 | -0.064 | 42.5% |
| K45 | 1110 | 6.92s | 0.11s | 0.22s | -0.030 | -0.084 | -0.354 | 83.4% |
| L3133 | 110 | 6m 39s | 0.12s | 1.65s | 0.215 | -0.173 | -0.616 | 53.3% |
| L3143 | 110 | >2h | 0.49s | 5.84s | 0.107 | -0.266 | -0.673 | 52.2% |
| L3151 | 110 | 2m 46s | 0.26s | 2.81s | -0.019 | -0.442 | -0.624 | 30.1% |
| L3153 | 110 | >2h | 0.99s | 8.91s | 0.024 | -0.363 | -0.681 | 45.1% |
| L3223 | 110 | 7m 44s | 0.20s | 12.25s | 0.516 | -0.525 | -1.299 | 42.6% |
| L3523 | 110 | >2h | 39.95s | 10m 39s | 0.953 | -1.953 | -6.671 | 61.9% |
| D33 | 110 | 3m 55s | 0.21s | 9.07s | 0.284 | 0.245 | 0.238 | 14.7% |
| D34 | 110 | >2h | 6.68s | 10m 14s | 0.284 | 0.208 | 0.139 | 47.7% |
| D62 | 111110 | >2h | 7.98s | 10m 6s | 0.333 | -0.000 | -0.000 | 0.0% |
| T350 | 101 | >2h | 0.10s | 1.09s | 0.600 | 0.509 | 0.352 | 63.3% |
| T3100 | 101 | >2h | 0.23s | 1.85s | 0.710 | 0.592 | 0.495 | 45.2% |

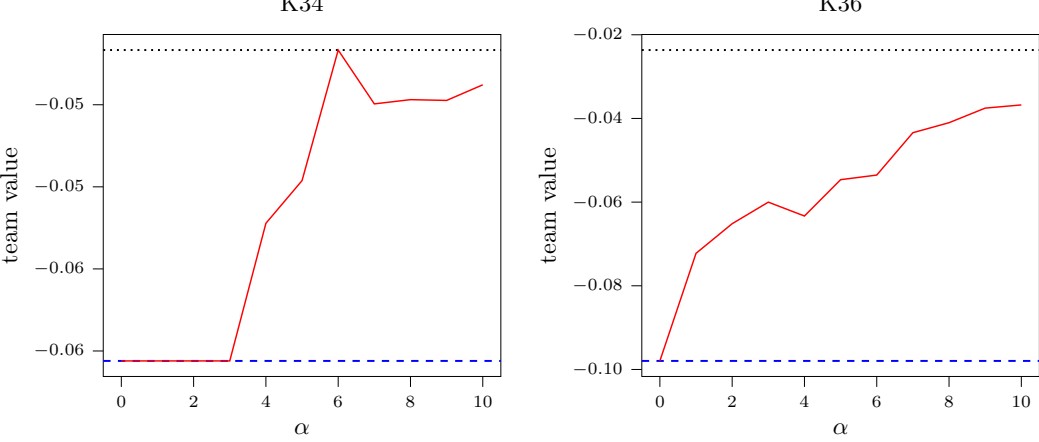

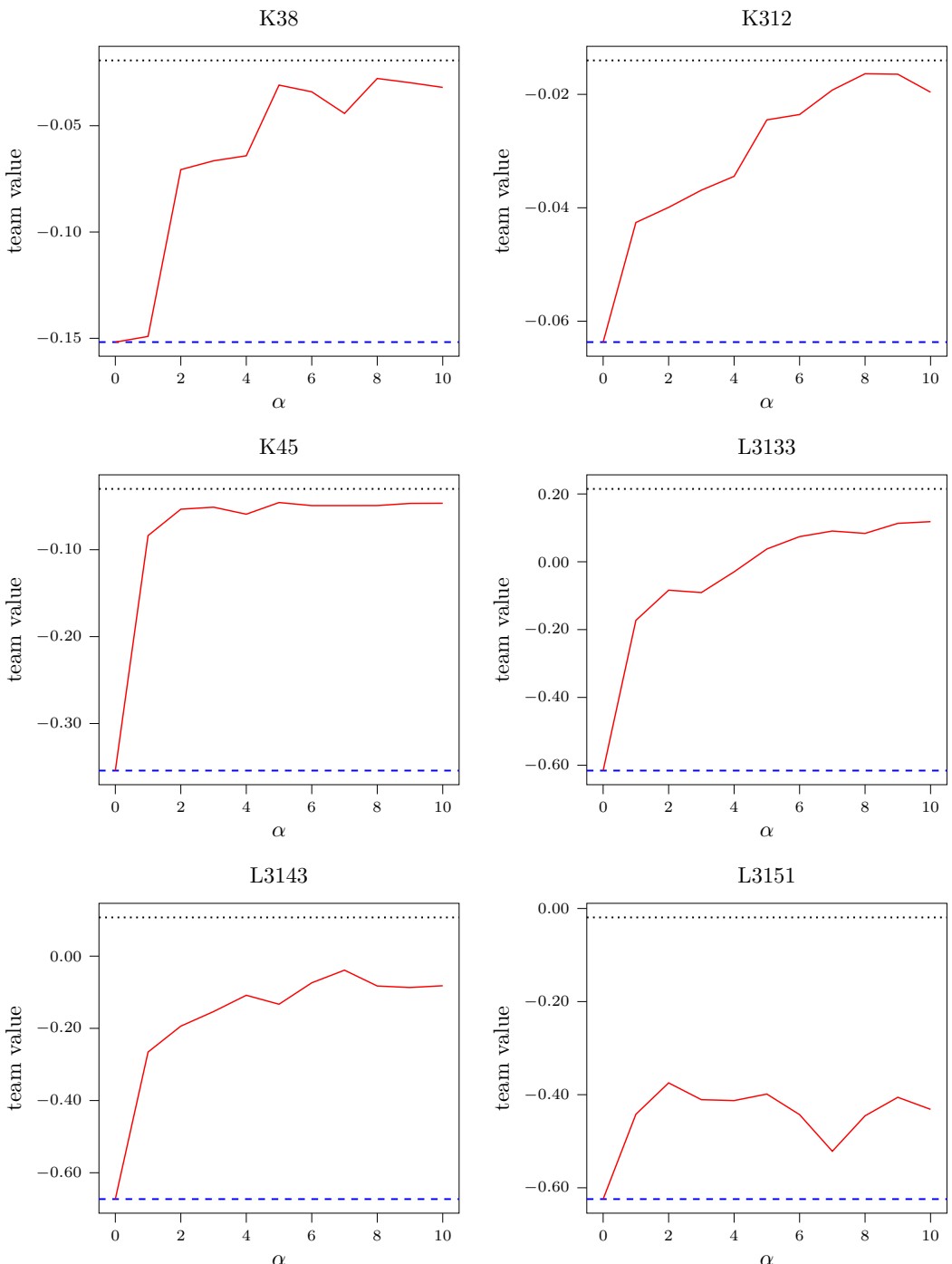

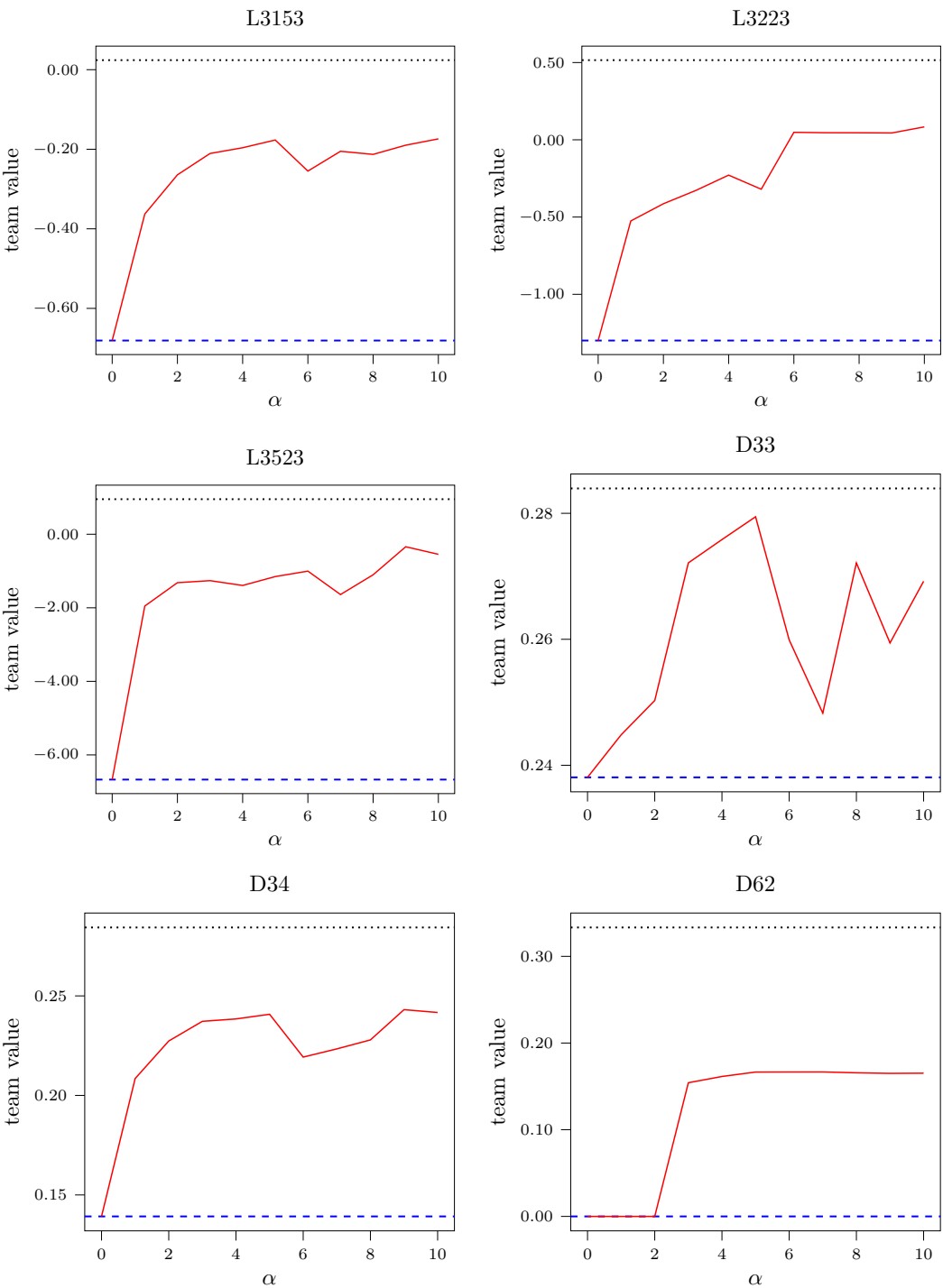

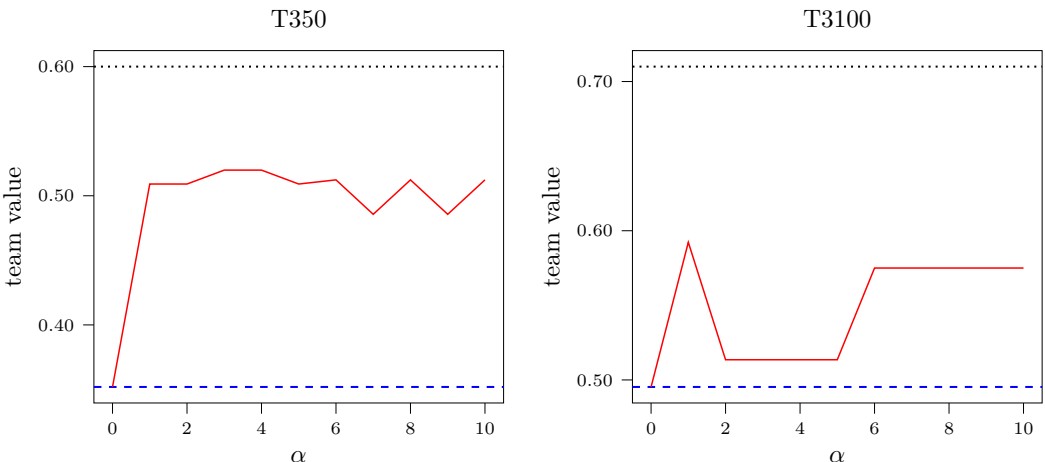

# D   Safety

**Theorem 5.** *Applying team maxmargin subgame solving (Definition 4) to every subgame reached during play, in a nested fashion, results in a playing a strategy with exploitability no worse than that of the blueprint.*

*Proof.* The blueprint has margin 0 by definition; therefore, the gadget subgame (Definition 4) always has nonnegative value. Moreover, if any subgame strategy $x'$ achieves nonnegative value in the gadget subgame, then the counterfactual best responses at the ▼-root beliefs cannot have improved for ▼ (by definition). Since the overall best response value for ▼ is a monotone function of these counterfactual best response values, the theorem follows. □

This is the same guarantee and argument given by Moravcík et al. [15] and Brown and Sandholm [1] in two-*player* games.