# OpenReview forum: "Subgame Solving in Adversarial Team Games"
_NeurIPS.cc/2022/Conference — NeurIPS 2022 Accept_

### Official Review · Reviewer_3Trb · 2022-06-25

**Rating:** 5
**Confidence:** 3
**Soundness:** 3 good
**Presentation:** 2 fair
**Contribution:** 2 fair

**Summary:**

This paper presents a subgame solving method that extends the existing approach to solving huge two-player zero-sum games. The main improvement of this paper is to solve the online optimization strategy through subgame solving. The subgame solving method is to generate the gadget game based on the team belief DAG (TB-DAG). The blueprint strategy is refined by iterative expansion using the column-generation (CG) algorithm, and its size increases linearly with the number of best-response oracle calls. That avoids the exponential size of the whole TB-DAG when expanding in the worst case. The blueprint generated by the CG algorithm is sparse. On this basis, the max-margin optimization problem in the subgame is converted into a linear program problem, which can be solved by the state-of-the-art integer program solver. Given the best-response oracle, the solving algorithm can be found to run end-to-end in polynomial time. Finally, they experimentally evaluated their method using a standard test suite and showed that this refinement of subgame solving is effective in reducing the difference between the value provided by the equilibrium strategy and the blueprint.

**Questions:**

1. How does the blueprint strategy generate? How to ensure that the blueprint strategy is a suboptimal strategy? Is it possible to generate sparse blueprint strategies if the game instances are relatively huge?
2. In Section 5.2, this paper mentions how the performance of this strategy refinement method varies as \alpha varies. Due to the nature of the CG algorithms, the improvement is sometimes not monotonically increasing in the number of iterations. What is the relationship between the performance of this method and \alpha? How about the performance when \alpha is greater than 5?
3. How does the equilibrium values calculate in the experiment?
4. The format of the conference names in the references is not uniform, some are abbreviations, some are full names, and the abbreviations of the names of the same conference are not consistent.
5. Figure 5.2, mentioned in Section 5.2, does not appear in the paper, there is only Figure 1.


**Limitations:**

The premise of the subgame solving approach proposed in the paper is that the blueprint strategies are assumed to be sparse, and the paper generates sparse blueprints by the CG algorithm. However, if the blueprint strategy generated by the CG algorithm is not sparse, the solution by the approach proposed in the paper may not be achieved in polynomial time.

**Strengths And Weaknesses:**

1. Originality
Compared with other adversarial team game approaches, the subgame solving approach proposed in this paper extends the maxmargin subgame solving to adversarial team games (ATGs), generates a gadget game from its TB-DAG subgame, uses the CG algorithm to generate a sparse blueprint and expand the blueprint strategy iteratively, given the best-response oracle, and enables the refinement of the blueprint strategy to approximate an exact solution.

2. Quality
The authors implement an online refinement strategy that approximates an exact solution without expanding the whole TB-DAG exponentially. However, there are the following problems in this paper.
(1) This paper lacks detailed theorems and proofs of the properties of approximation to the exact solution.
(2) There are two ways to generate sparse blueprints, but the experimental evaluation only considers the sparse blueprints generated by the CG algorithm, and not on the sparse blueprints obtained by the other method, i.e., taking an arbitrary blueprint in realization form and expressing it as a sparse convex combination by Caratheodory’s theorem. The experimental evaluation is not complete and sufficient.

3. Clarity
The paper is somehow well-structured and the language is fluent.
(1) The paper links to the literature giving relevant game examples, but the overall process of specific solving using the approach proposed in this paper is not intuitive and clear enough.
(2) In the experimental results, the meaning of each index in Table 1 is not explained one by one.

4. Significance
The subgame solving technique proposed in this paper is effective for reducing the difference between blueprint strategy and equilibrium strategy and extends the successful approach of solving huge two-player zero-sum games, in which the CG algorithm is a very important component of subgame solving technology. Methods similar to column generation may also be used as fundamental building blocks in solving team games. However, in the experimental results, the team structure size of the game instances given is small.

---

> ### Author Response · Authors · 2022-08-02
> **Response to review**
>
> *Detailed theorems and proofs*
>
> We state and prove the usual properties of subgame solving, namely safety, in Appendix D.
>
> *Experiments using the Caratheodory-based method*
>
> We noted the Caratheodory-based method for achieving a sparse blueprint mostly for theoretical interest: if one had an alternative method of generating a blueprint, Caratheodory can be used to sparsify it. However, on many games we test on, methods based on CG (or its neural equivalent, PSRO) are the only known practical methods of generating a blueprint (see experimental results in [20]). Further, the support size of CG is bounded by the number of iterations for which it runs (see line 277), which is usually far less than |Z| and therefore would beat such a Caratheodory-based method. For these reasons, our experimental results focus on CG. We will explicitly point this out in the final version.
>
> *Clarity on "overall process of specific solving" using the approach*
>
> Our paper is the first on subgame solving in ATGs, and follows the usual framework taken by several other previous papers about subgame solving in two-player zero-sum games (e.g., [1, 3, 4]). We will add some background information about subgame solving in the final version.
>
> *Columns of Table 1*
>
> Game instance and team structure are explained in Section 5.1. Equilibrium time is the time it took for CG to compute an exact equilibrium. Refinement time is the time allotted to the subgame solving algorithm per each resolving step. Blueprint time is the time allotted to the blueprint computation. Equilibrium value is the equilibrium value of the game. Refinement value is the value of the refined strategy against a best response. Blueprint value is the value of the blueprint against a best response. Gap reduction is defined in Section 5.2. We will add these column-by-column descriptions to Table 1’s caption.
>
> *Small size of experiments*
>
> We focus on small games in experiments so that we can compute exact game values, exploitability values, etc. See also response to Reviewer iqNg.
>
> *How does the blueprint strategy generate? How to ensure that the blueprint strategy is a suboptimal strategy? Is it possible to generate sparse blueprint strategies if the game instances are relatively huge?*
>
> In large games where subgame solving is usually applied, the sheer size of the game prevents the offline computation of an optimal strategy, so subgame solving papers assume by default that the blueprint is not optimal. Of course, if the blueprint were already optimal, then subgame solving would not be useful. In our experiments, the blueprint strategies are computed by early stopping, as specified in Line 302, and we make sure they are not optimal by evaluating the value achieved against a best response.
>
> In case the game is too large, blueprints can be computed by using more scalable techniques (such a PSRO) or by solving an abstracted version of the game, as customarily done to solve large game instances. Column generation (in particular, its neural form, which is PSRO) is one of the few techniques that is known to be scalable to huge games in the two-player zero-sum setting. We hope that our paper will therefore shed light in the future on how to correctly apply PSRO to ATGs, which would be a large step toward human-level or superhuman performance in this setting.
>
> Regarding the sparsity of the generated blueprint, this is guaranteed when CG/PSRO is used as the underlying game-solving algorithm, since the number of pure strategies belonging to its support is bounded by the number of iterations that are run (and empirical results show that it is much sparser in practice)
>
> *Performance dependence on $\alpha$*
>
> Performance generally improves with larger $\alpha$, though obviously with diminishing returns, and some nonmonotonicity as the reviewer correctly points out. This is clearly exhibited in the plots in the Appendix C. Although we did not test it, we believe that this general behavior will continue past $\alpha = 5$.
>
> *How does the equilibrium values calculate in the experiment?*
>
> By running CG to completion, or by taking the equilibrium values from [20] in case CG did not finish within the time limit. (See also the caption of Table 1.)
>
> *If the blueprint strategy generated by the CG algorithm is not sparse, the solution by the approach proposed in the paper may not be achieved in polynomial time.*
>
> This is true. However, as we pointed out (see answer to “Quality” point 2, above), CG/PSRO are the only current candidate methods that may scale, and they are sparse.

---

> > ### Comment · Reviewer_3Trb · 2022-08-09
> > **After reading the response**
> >
> > Comment:
> > Thanks for the through response!
> >
> > Detailed theorems and proofs: The theorems and proofs given in Appendix D illustrate that strategies with exploitability no less than blueprints can be achieved through the team maxmargin subgame solving. But it seems that the formal exact solution approximation property would be clearer.
> >
> > Experiments using the Caratheodory-based method: It would be clearer if it could be pointed out that using CG to generate blueprints is preferred over the Caratheodory-based method.
> >
> > Clarity on “overall process of specific solving” using the approach: I feel it is not only necessary to add the background information related to the subgame solving in two-player zero-sum games, but also to give a specific actual solving process using the method of the paper, which will make the key points of the paper clearer.
> >
> > Columns of Table1: It would make the experimental data more understandable if a column-by-column description could be added to the Table 1’s caption.
> >
> > About blueprint strategy: Thanks for the clarification that for large games it is still possible to compute sparse blueprints by using column generation techniques or by solving an abstract version of the game. But the game instances in the experiment are relatively small, and I think it would be better to add some experimental explanations for solving large game instances.
> >
> > Performance dependence on \alpha: I see what you mean, but due to its non-monotonicity, I feel that testing the performance of the strategy refinement method when \alpha is greater than 5 would make the paper more rigorous.
> >
> > Equilibrium values: Thanks for your explanation, I roughly understand how the equilibrium value is calculated.
> >
> > Limitations: To some extent, I agree with you, but the basis of what the proposed method can achieve in polynomial time is indeed that the generated blueprint strategy must be sparse, which is still rather limiting.

---

> > > ### Author Response · Authors · 2022-08-09
> > > **Reply**
> > >
> > > **Detailed theorems and proofs: The theorems and proofs given in Appendix D illustrate that strategies with exploitability no less than blueprints can be achieved through the team maxmargin subgame solving. But it seems that the formal exact solution approximation property would be clearer.**
> > >
> > > Assuming that “formal exact solution approximation property” you mean that you want a proof that the algorithm finds a TMECor: our paper is about subgame solving, and subgame solving techniques generally do not converge to exact equilibrium; rather, the goal is improvement over a blueprint. Please refer to the most recent response to iqNG for more.
> > >
> > > **Clarity on “overall process of specific solving” using the approach: I feel it is not only necessary to add the background information related to the subgame solving in two-player zero-sum games, but also to give a specific actual solving process using the method of the paper, which will make the key points of the paper clearer.**
> > >
> > > By “overall process of specific solving” do you mean that you want an explicit algorithm finds a TMECor via subgame solving? If this is the case, please refer to the previous answer.
> > >
> > > If instead by “overall process of specific solving” you refer to a more intuitive explanation of team maxmargin as we have done for the original maxmargin algorithm in Line 162-169, then we can add some extra explanations at the end of section 3.
> > >
> > > **Experiments using the Caratheodory-based method: It would be clearer if it could be pointed out that using CG to generate blueprints is preferred over the Caratheodory-based method.**
> > >
> > > We will add some text after Line 279 to be more precise on this topic.
> > >
> > > **Columns of Table1: It would make the experimental data more understandable if a column-by-column description could be added to the Table 1’s caption.**
> > >
> > > Thanks for the suggestion, we will use our previous response to expand Table 1’s caption.
> > >
> > > **About blueprint strategy: Thanks for the clarification that for large games it is still possible to compute sparse blueprints by using column generation techniques or by solving an abstract version of the game. But the game instances in the experiment are relatively small, and I think it would be better to add some experimental explanations for solving large game instances.**
> > >
> > > We agree on the need for experimental results with huge games. However, let us remark that, with these games, we cannot compute the equilibrium, and therefore we cannot evaluate the gap reduction our techniques guarantee. The only possibility is to show the improvement in terms of expected utility, and in large games such an evaluation can only be performed through MonteCarlo estimation. We leave this exploration for future research, as the current paper serves to introduce the idea of subgame solving in ATGs.
> > >
> > > **Performance dependence on \alpha: I see what you mean, but due to its non-monotonicity, I feel that testing the performance of the strategy refinement method when \alpha is greater than 5 would make the paper more rigorous.**
> > >
> > > We will increase alpha further in the next version and see what happens.

---

### Official Review · Reviewer_a16J · 2022-07-02

**Rating:** 6
**Confidence:** 5
**Soundness:** 4 excellent
**Presentation:** 3 good
**Contribution:** 3 good

**Summary:**

The paper proposes a modification of subgame refinement for adversarial team games. An abstract strategy is refined for the public states encountered in the game with additional iterations using a modification of the max-margin gadget. The empirical evaluation on multiple games that can completely fit into memory shows that most of the gap from the optimal solution can be removed by this method.

**Questions:**

Are the results transferable to general imperfect recall games?

**Limitations:**

Limitations were addressed IMO adequately and I do not see negative societal impact of the work.

**Strengths And Weaknesses:**

Decomposition on solving large imperfect information games in a relevant problem and this paper makes an incremental, but solid step in generality of this approach. I am not aware of any other work on subgame refining in team games, or even imperfect recall games in general. The paper is well written, reasonably easy to follow, and I did not find any serious technical mistakes.

The experiments are sufficiently thorough for a conference paper. They show a clear merit of the approach. On the other hand, they fail to explore more widely the practical applicability tradeoffs. How much larger games are now solvable with this approach? One of the key complications of the presented approach is the use of DAG strategy representation, which is certainly important if the game is supposed to fit into the memory. Subgame solving is likely to push the boundary a little, but combining it with depth-limited solving would most likely be much more impactful. With a value funcion the difference between DAG and tree representation would become much less relevant.

Team games are becoming an established line of work. However, as the authors admit themselves, it is a special case of imperfect recall games and it is not clear to me why to study them separately. Is the subgame solving in this paper applicable also in general imperfect recall games or are there some assumptions given by the special case that are necessary?

Minor issues:

 Line 53: It seems to claim that [15] was using an offline computed blueprint strategy, which is incorrect.

Line 277: A little more details on which Caratheodory’s theorem you mean and how to use it would be appreciated.

I assume that the refinement is run from each decision node in the experiments, but I did not find it explicitly stated anywhere.

---

> ### Author Response · Authors · 2022-08-02
> **Response to review**
>
> *“Line 53: It seems to claim that [15] was using an offline computed blueprint strategy, which is incorrect.”*
>
> Thanks for catching this; we have corrected it.
>
> *"Line 277: A little more details on which Caratheodory’s theorem you mean and how to use it would be appreciated.”*
>
> Sorry for the confusion. We were referring to the [theorem that pertains to convex analysis](https://en.wikipedia.org/wiki/Carath%C3%A9odory%27s_theorem_(convex_hull)). The theorem states that any convex combination of points in R^d can be expressed as a convex combination of at most d+1 of them. We have added a footnote stating this. In our specific case, this implies that any strategy for a team can be expressed with a support of at most |Z| team pure strategies.
>
> *“I assume that the refinement is run from each decision node in the experiments, but I did not find it explicitly stated anywhere.”*
>
> Refinement is performed at every public state. We have explicitly stated this in the new version.
>
> *Are the results transferable to general imperfect recall games?*
>
> Yes. Two-player zero-sum imperfect-recall games are equivalent to two-team zero-sum games, so our results transfer immediately to that setting.

---

### Official Review · Reviewer_iqNg · 2022-07-06

**Rating:** 3
**Confidence:** 4
**Soundness:** 3 good
**Presentation:** 3 good
**Contribution:** 2 fair

**Summary:**

This paper focuses on solving large adversarial team games, which include both cooperation and competition. They extend the subgame solving technique of solving huge two-player zero-sum games, which computes a blueprint strategy offline and refines it online. The proposed approach is based on team-belief DAG for the gadget game and column generation algorithm for refining the blueprint strategy. They empirically evaluate the proposed approach to see if the blue strategy is improved.

**Questions:**

No

**Limitations:**

Yes

**Strengths And Weaknesses:**

Existing algorithms are hard to solve large games, e.g., the performance of (column generation) CG is constrained by the requirement of memory and iterations. To overcome this challenge, this paper extends the subgame-solving technique to AGT by handling its unique property in AGT.
However, this proposed subgame-solving technique still uses CG for both the blueprint and the subgame-solving procedure. Then this proposed algorithm may still suffer the limitation of existing algorithms.

This work aims to solve large games, but games used in experiments are very small, and all of them can be solved efficiently by existing algorithms, e.g., algorithms in [20]. Even in these small games, e.g., Leduc poker games, the gap between the equilibrium value and the refinement value is still very large. If this proposed solving algorithm cannot perform well in these small games, it is hard to imply that it can perform well in larger games.

It is interesting to see the performance of the algorithms in [20] or [21] using the same time as the subgame solving algorithm

The following multiagent learning-based algorithm for TMECor that is not limited by memory should be a baseline.\
Cacciamani F, Celli A, Ciccone M, Gatti N. Multi-Agent Coordination in Adversarial Environments through Signal Mediated Strategies. In Proceedings of the 20th International Conference on Autonomous Agents and MultiAgent Systems 2021 May 3 (pp. 269-278).

Line 3 of Algorithm 1: ‘alt’ ->’all’

---

> ### Author Response · Authors · 2022-08-02
> **Response to review**
>
> *“[..] games used in experiments are very small, and all of them can be solved efficiently by existing algorithms, e.g., algorithms in [20]. Even in these small games, e.g., Leduc poker games, the gap between the equilibrium value and the refinement value is still very large. If this proposed solving algorithm cannot perform well in these small games, it is hard to imply that it can perform well in larger games.
> It is interesting to see the performance of the algorithms in [20] or [21] using the same time as the subgame solving algorithm”*
>
> We believe that comparing exact algorithms (e.g., those in [20]) with our paper is not apples to apples. While [20] provides algorithms for solving games from scratch, our paper provides a method for refining blueprints (i.e., subgame solving), a different task that is orthogonal to [20] (for instance, one could use [20] to compute a blueprint and then use subgame solving to refine it). In the past, subgame solving has proven to be a key technology for constructing superhuman AIs for games. In this paper, we show for the first time that subgame solving technology can be extended to adversarial team games, despite the many additional technical challenges associated with those settings. Testing the scalability of subgame solving technology applied to different blueprint strategies is an interesting direction of future research. For this paper, we focused on experimentally validating the algorithm on small games that are solvable exactly in reasonable time (see e.g. [20] for timing results), so that we can compute exploitability values and therefore exact blueprint refinement percentages—see also reviewer a16J’s review of experimental evaluation. In other words, we did it so we can rigorously evaluate the techniques. In larger games where best response computation is not computationally feasible, such evaluation cannot be carried out. Our results confirm that the blueprint refinement technology appears very promising experimentally even in adversarial team games. To summarize: the experimental timings are not intended to be directly compared to blueprint-producing algorithms; that is not the purpose of this paper.
>
>
> *“To overcome this challenge, this paper extends the subgame-solving technique to AGT by handling its unique property in AGT. However, this proposed subgame-solving technique still uses CG for both the blueprint and the subgame-solving procedure. Then this proposed algorithm may still suffer the limitation of existing algorithms.”*
>
> The central question we aim to answer in this paper is whether the local reoptimization performed by subgame solving provides benefit. In doing that, we opted to use the most theoretically grounded algorithm, so that our algorithms and experimental evaluation are concrete. Since the answer is positive (i.e., subgame solving does provide benefit), the next question is how to scale up such a technique.
> In this perspective, we agree that the algorithm, as written in the paper, may suffer from the limitations of CG because it uses CG as a subroutine. Suffering some form of exponential blowup is unavoidable due to NP-hardness lower bounds (see [6]). So, this limitation is to a large extent inherent to the problem.
> Furthermore, column generation (in particular, its neural form, which is PSRO) is one of the few techniques that is known to be scalable to huge games in the two-player zero-sum setting. We hope that our paper will therefore shed light in the future on how to correctly apply PSRO to ATGs, which would be a large step toward human-level or superhuman performance in this setting. See also the response to Reviewer DBNN above, where we give an overview of the technical contributions of the paper.
>
>
> *“The following multiagent learning-based algorithm for TMECor that is not limited by memory should be a baseline.
> Cacciamani F, Celli A, Ciccone M, Gatti N. Multi-Agent Coordination in Adversarial Environments through Signal Mediated Strategies. In Proceedings of the 20th International Conference on Autonomous Agents and MultiAgent Systems 2021 May 3 (pp. 269-278).”*
>
> We thank the reviewer for the suggestion. The method proposed in the cited work guarantees convergence to a TMECor **only in the special case in which the team players have symmetric observability** (e.g., either all the players on the same team observe something or none of them do). In our paper, instead, we consider general team games, in which the players are allowed to have private information (for instance the private cards in a card game). In such settings, the suggested baseline is not applicable.

---

> > ### Comment · Reviewer_iqNg · 2022-08-09
> > **After response**
> >
> > The response could not convince me.
> >
> > About the baseline, authors mentioned the cited baseline in my review may not converge to a TMECor in some games, which does not mean it cannot work well in practices, as shown in their experiments. Moreover, the proposed method does not converge even in small games due to the large gap shown in experiments, which makes me hard to believe it can guarantee convergence to a TMECor, especially, in large games.

---

> > > ### Author Response · Authors · 2022-08-09
> > > **On the purpose of subgame solving and the experiments**
> > >
> > > **Moreover, the proposed method does not converge even in small games due to the large gap shown in experiments, which makes me hard to believe it can guarantee convergence to a TMECor, especially, in large games.**
> > >
> > > Guaranteeing convergence to TMECor is not, cannot, and should not be the goal of subgame solving. This is true also in the two-player zero-sum setting, it is true of all prior techniques for subgame solving in imperfect-information games, and it is true for us. Instead, the goal is to be able to play *better than the blueprint you are given*. If the blueprint is suboptimal, so will the final strategy returned by subgame solving. However, subgame solving will do better than the blueprint. That is what our experiments attempt to demonstrate. It is why we intentionally use suboptimal strategies as our blueprints (it is pointless to perform subgame solving on an already-optimal strategy), and intentionally (quite severely!) limit the amount of computation allocated to the subgame solve: we want to show that, even with very limited compute, subgame solving in team games can improve a blueprint. We will clarify this in the next version.
> > >
> > > **About the baseline, authors mentioned the cited baseline in my review may not converge to a TMECor in some games, which does not mean it cannot work well in practices, as shown in their experiments.**
> > >
> > > That paper does not run experiments on any game that does not have symmetric observability. But, as we argue in the previous paragraph, we believe that the discussion of this baseline is orthogonal to the point of our paper, since our paper is not about guaranteed convergence to TMECor.

---

> > > > ### Comment · Reviewer_iqNg · 2022-08-10
> > > > **RE**
> > > >
> > > > My point is that the above work I mentioned should be an important baseline for subgame solving because both cannot guarantee convergence to TMECor. The current version of subgame solving still suffers the limitation of memory due to the CG subroutine, but the above work I mentioned does not suffer that limitation. Showing results in large games is also important for evaluating subgame solving. So I think the current version of the paper fails to show the advantage of subgame solving on computing TMECor.

---

### Official Review · Reviewer_DBNN · 2022-07-21

**Rating:** 6
**Confidence:** 2
**Soundness:** 4 excellent
**Presentation:** 3 good
**Contribution:** 2 fair

**Summary:**

The paper addresses approximating equilibrium strategies in adversarial team games (ATGs).  The paper claims the following contributions: (1) It is “the first method for refining strategies by subgame solving in ATGs,” which it does by generating a “blueprint strategy” using a column-generation algorithm (with a time limit).  (2) it develops the Team-maxmargin algorithm, which is motivated by the existing maxmargin algorithm.  This algorithm in experimental evaluations “reduces the gap between the values provided by the equilibrium strategy and the blueprint.”


**Questions:**

I'd be happy for the authors to correct anything in my review that is incorrect or denotes a misunderstanding of the paper's contributions.

**Limitations:**

Please see the previous sections

**Strengths And Weaknesses:**

Up-front comment so that my review can be taken in proper context: This paper is in the rather niche area of solving zero-sum games, an area that has been developed extensively over the last decade or two primarily by a handful of research groups.  While being familiar with that area, I am not immersed in it, nor have I followed the notation, terminology, and advancements extremely closely.  Typically, however, I can follow papers in this area.  But I had difficulty following the details of the paper and its high-level message.   I followed the terminology and notation well for a while as I read into the paper, but then the overwhelming number of terms, etc. caused me to begin to lose context (and hence interest).  That said, during my efforts, I noted that the paper is written cleanly and carefully, and appears to be technically sound.  The paper exhibits a high amount of technical knowledge, I just couldn’t follow it the whole paper through.  On the downside, I also failed to grasp the key technical messages of the paper (what makes it tick? What knowledge generalizes?) within the time-limits that I had (being an emergency reviewer for this paper).

With that context in mind, here is my evaluation of the contributions of the paper:

(1) Is the claim that the paper is the first to present “a method for refining strategies by subgame solving in ATGs” substantiated in the paper?  No.  I recommend that this claim be removed from the paper.  First, it would be impossible to substantiate.  Second, the contribution of the paper likely wouldn’t be reduced even if the claim was proven to be false.  Third, the claim hurts the paper in that it focuses the reader on whether it really is the first rather than the actual content of the paper.  Thus, I think the paper would be better if the claim were removed.

(2) While I found the team-maxmargin algorithm difficult to follow, the empirical results across four different zero-sum games do appear to indicate that it does reduce the gap between the values provided by the equilibrium strategy and the blue print.  I find it nice that the paper considers the four games.

Is the algorithm novel?  I believe it is.  The knowledge development is somewhat incremental in that the approach builds well upon existing work in the well-developed area.

Is the contributed knowledge important?  It depends on how much one values game playing.  It certainly is a fun topic, though one could argue that such game-playing is unlikely to lead to useful applications in the near future.  Even still, it is possible that knowledge generated in this domain could become useful outside of the niche area.  And this paper represents an impressive effort that can expand knowledge for those in the area.

---

> ### Author Response · Authors · 2022-08-02
> **Response to review**
>
> *“I also failed to grasp the key technical messages of the paper (what makes it tick? What knowledge generalizes?) within the time-limits that I had (being an emergency reviewer for this paper).”*
>
> Thanks for serving as an emergency reviewer for the paper and for being so upfront. The central technical contribution of the paper is to establish that subgame solving in the context of ATGs is possible (this has never been done before) and that it provides non-negligible benefits. This is not obvious, given the intrinsic complexity that comes with team games (which is reflected, for example, in various negative computational complexity results that are unique to two-team zero-sum games and do not apply to two-player zero-sum games). Given the fundamental role that subgame solving has played in the construction of superhuman AIs for games (e.g., chess, Go, and poker; see also paragraph at line 50), we are confident that extending the framework of subgame solving to the more challenging setting of ATGs will be an important stepping stone towards superhuman AIs for ATGs.
>
> Specific technical contributions in the paper to achieve this include:
>
> 1. as detailed in line 233, the auxiliary game to solve while performing subgame solving cannot be explicitly represented as a game on its own (this is a major difference from the two-player zero-sum case), since reach probabilities depend on the adversary team belief. Such a constraint cannot be represented explicitly through some game rules, and the correct representation of such a constraint was a challenging step from a theoretical point of view.
> 2. we highlighted the special synergy between column generation methods and subgame solving techniques in the specific setting of team games. This stems from the huge benefit of sparsity when dealing with a possibly exponential number of possible team beliefs.
>
> *“Is the claim that the paper is the first to present “a method for refining strategies by subgame solving in ATGs” substantiated in the paper?”*
>
> We are not aware of any other paper discussing subgame solving in adversarial team games, so to our knowledge we are the first to present a method for subgame solving in ATGs (if the reviewer is aware of such a paper, we are happy to include a comparison in the final version). Given how important search has been in zero-sum two-player games (see paragraph at line 50), we believe that extending the framework of subgame solving to the more challenging setting of ATGs is very significant.

---

### Author Response · Authors · 2022-08-02
**Note to all reviewers**

As already noted in the previous version’s supplementary material, between the submission and supplement deadlines, we discovered an error in the reporting of the experimental results, in which, due to a coding mistake, the reported blueprint value was actually the value of a random strategy, not the blueprint. The supplemental materials submitted with the original version of the paper contained a corrected experimental table and plots. While the gap reduction values are now slightly lower, the qualitative interpretation and conclusions we draw from the experimental results remain unchanged. We apologize for any inconvenience. The new revision we have uploaded fixed the mistake.

---

### Meta-Review · Area_Chair_BDNK · 2022-08-27

**Recommendation:** Accept
**Confidence:** Less certain

**Metareview:**

The most serious and focused criticism for this paper comes from a single reviewer, who also happens to be the most confident reviewer. The crux of the criticism is that the authors should have compared against the 2021 AAMAS paper, "Multi-Agent Coordination in Adversarial Environments through Signal Mediated Strategies" and the algorithm described therein. The authors point out that this algorithm as designed for symmetric observations and was never tested in the more general case. The reviewer thinks it might work well in the asymmetric case. He could be right, but it seems a bit much to require the authors compare with a reviewers hunch about modifications to existing work. If we allow reviewers to drag authors down that rabbit hole, nothing might ever get published.

The sentiment of the other reviewers is weakly positive. There is general concern that the games used in the experiments are not particularly large, and overall concerns about scaling to larger games.

On the positive side, the approach does appear to be novel and it does appear to do well in the provided experiments.

**Award:**

No

---

### Decision · Program_Chairs · 2022-09-14

Accept